# Artificial colloidal liquid metacrystals by shearing microlithography

Yanqiu Jiang[1,2], Fan Guo[1,2], Zhen Xu [1], Weiwei Gao[1] & Chao Gao[1]

Meta-periodicity beyond intrinsic atomic and molecular order, such as metacrystalline and quasicrystalline lattices, exists in solids, but is usually elusive in lyotropic liquid crystals for its energetic instability. The stable meta-periodicity in lyotropic liquid crystals in the absence of external stimuli remains unexplored, and how to achieve it keeps a great challenge. Here we create lyotropic liquid crystals with stable meta-periodicity in a free state, coined as liquid metacrystals, in colloidal systems by an invented shearing microlithography. The meta-periodicity is dynamically stabilized by the giant molecular size and strong excluded volume repulsion. Liquid metacrystals are designed to completely cover a library of symmetries, including five Bravais and six quasicrystalline lattices. Liquid metacrystal promises an extended form of liquid crystals with rich meta-periodicity and the shearing microlithography emerges as a facile technology to fabricate liquid meta-structures and metamaterials, enabling the digital design of structures and functionalities of liquid crystalline materials.

[1] MOE Key Laboratory of Macromolecular Synthesis and Functionalization, Department of Polymer Science and Engineering, Key Laboratory of Adsorption and Separation Materials & Technologies of Zhejiang Province, Zhejiang University, 38 Zheda Road, 310027 Hangzhou, P. R. China. [2]These authors contributed equally: Yanqiu Jiang, Fan Guo. Correspondence and requests for materials should be addressed to Z.X. (email: zhenxu@zju.edu.cn) or to C.G. (email: chaogao@zju.edu.cn)

**M**eta-periodicity, such as metacrystalline and quasicrystalline lattices exists in solids, but is usually elusive in fluids since the weak intermolecular interaction in liquid is feeble to provide the requisite energy[1]. Simple liquids are disordered and ordinary liquid crystals (LCs) only feature randomly distributed orientational order of anisotropic mesogens[2–9]. In LCs, anisotropic mesogens spontaneously organize into a monotonic orientational order, described by the preferred orientation of molecular director ($\hat{n}$)[1,2]. Modulating $\hat{n}$ into spatial meta-periodicity produces as-demand structures and functions in daily displays[10], adaptive optics[11], lasers[12], structural fibers[13], and elastomers[14]. However, meta-periodicity of $\hat{n}$ appears as curvature strain and induces strong elastic stress, which increases the elastic free energy of LCs and behaves thermodynamically unstable[2].

To stabilize meta-periodicity of $\hat{n}$, continuous external stimuli are usually required, such as electrical[6–8] or magnetic[9] fields, light or thermal stimuli[15,16], and permanent surface anchoring[3–5]. For small molecular LCs, the meta-periodicity in fluid state tends to dynamically fade away after a short relaxation time ($\tau$, milliseconds) for the fast random rotation of small mesogens under thermal fluctuation[17]. Alternatively, the stability of meta-periodicity could be enhanced in highly viscous systems[18–21]. For instance, Coles et al. created single texture in theromotropic smectic LC of polysiloxane by electric field-assisted thermal laser writing[18]. However, for lyotropic LCs with low viscosity, how to achieve stable meta-periodicity without external stimuli still remains a challenge.

Giant mesogens with aspect ratio ($\alpha$), such as 2D materials[13], amphiphilic polymers[15], and nanoparticles[6], can form lyotropic LCs at low volume fraction due to their prominent structural anisotropy. Accompanying with the giant $\alpha$ is the large rotation energy barrier of mesogens, which can increase the relaxation activation energy and dramatically lengthen $\tau$[22]. Therefore, meta-periodicity of $\hat{n}$ in a free state might be kinetically arrested in colloidal lyotropic LCs with giant molecular size, although the enormous energy demand caused by giant $\alpha$ disables conventional electromagnetic method to manipulate $\hat{n}$[3–9,15–17].

In this work, we fabricated lyotropic liquid metacrystals (LMCs) in colloidal systems by a shearing microlithography (SML) technique. The giant molecular size of colloids and strong excluded volume repulsion in colloidal LCs synergistically retard the relaxation of programed mesogens and dynamically arrest meta-periodicity with high stability for months. The programmable SML conquer the enormous energy barrier and construct LMCs with desired periodicity. LMCs are designed to completely cover a library of symmetries, ranging from Bravais lattices and quasicrystalline lattices in 2D space to Moiré lattice and tetrahedron in 3D space. The spatial meta-periodicity in LMCs fades following the Avrami kinetics of solids, and anisotropically melts at a specific vibration frequency. In addition, LMCs are converted to solid materials with designed periodicity and enable the digital design of overall properties of LC based materials.

## Results

**Relaxation behaviors of graphene oxide liquid crystals.** Graphene oxide (GO) aqueous solution was utilized as a model system ($\alpha$ of GO sheets ~15000, Supplementary Fig. 1) to realize LMC since single-layer GO is largely available and can easily form stable lyotropic mesophases[13]. Figure 1a–c shows its transition from isotropic phase (I) with chaotic distribution to nematic phase (N) with monotonically aligned $\hat{n}$ upon $c$[1,2,13,22]. Tracking single shearing induced grain, two topological defects with strengths of $+1/2$ (head) and $-1/2$ (tail) annihilated after $\tau$ (Fig. 1d, e, Supplementary Fig. 2), behaving as opposite charges[23]. $\tau$ generally increases with $c$ following an power law as $\tau \propto c^\rho$[24,25], and $\rho$ index sharply grows from 1.3 in I to a colossal value of 6.7 in N phase, transiting at a critical concentration ($c^*$) around 0.08 wt.%.

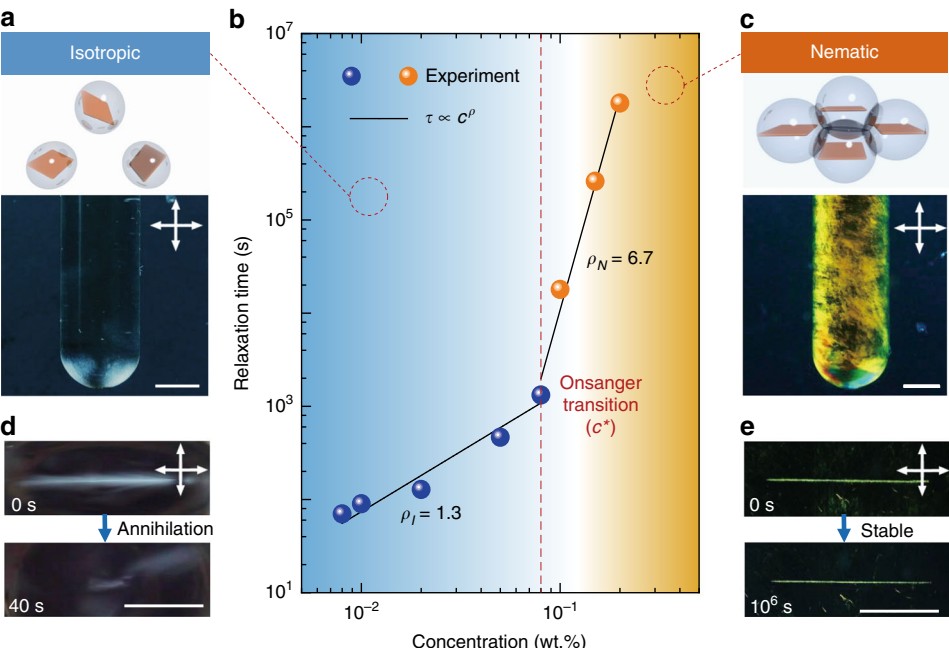

**Fig. 1** Relaxation diagram of graphene oxide liquid crystals. **a, c** Schematic illustrations of GO sheets distribution and corresponding POM images of I (**a**) and N phase (**c**), scale bars, 5 mm. **b**, $\tau$ of annihilation of re-orientated grain as a function of $c$. Dots are measured $\tau$ and solid lines are fittings by $\tau = Bc^\rho$ (See Methods). The red dash line indicates the $\tau$ transition, synchronous with I-N transition. **d, e** POM images for a tracking grain in I phase (**d**), which completely relaxes in a short time (~40 s), and a grain in N phase (**e**) with negligible relaxation after $10^6$ s, scale bars, 1 mm

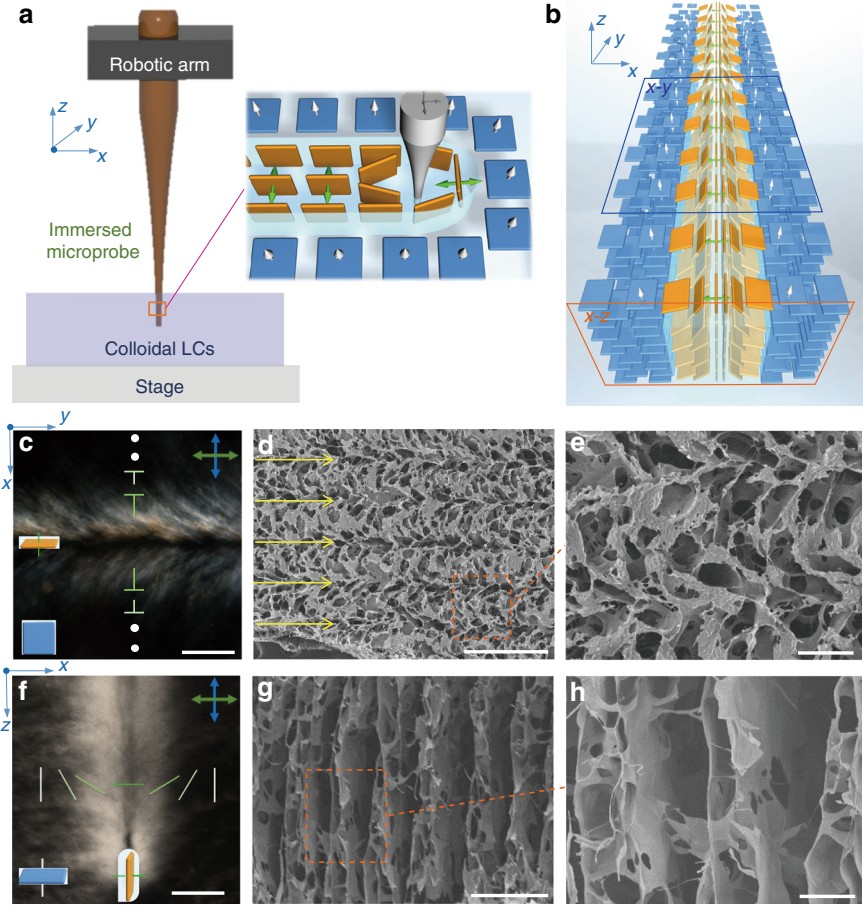

**Fig. 2** Shearing microlithography to modulate $\hat{\mathbf{n}}$. **a** Schematic of SML. The programmable moving probe is immersed into GO LCs to generate localized shearing field to vertically arrange GO sheets along shearing. **b** A 3D structural model of a π wall created by SML. GO sheets gradually turn to a vertical configuration in shearing center and $\hat{\mathbf{n}}$ undergoes a 180° rotation across the domain. **c**–**h** Top view (**c**–**e**, in x–y plane) and section view (**f**–**h** in x–z plane) of the reoriented domain under POM (**c**, **f**) and SEM (**d**, **e**, **g**, **h**), scale bar, **c**, **f** 20 μm; **d**, **g** 200 μm; **e**, **h** 50 μm

The τ transition is synchronous with the I-N phase transition (Fig. 1b), and $c^*$ proximately coincides to the phase transition concentration (0.066 wt.%) predicted by Onsager's excluded volume theory (Methods)[22]. In I phase, GO sheets randomly distribute and independently oscillate near their equilibrium sites following a Brownian rotation behavior (Fig. 1a)[26]. The increasing $c$ shortens intermolecular distance and enhances repulsive interaction, thus elevating the rotation energy barrier. In N phase, the rotation of one sheet alters its neighbors by excluded volume overlapping (Fig. 1c), dramatically enlarging the rotation energy barrier. As a result, ρ jumps from 1.3 (in I) into 6.7 (in N). We alternatively evaluated the relaxation rate by assessing "$\hat{\mathbf{n}}$ diffusivity" ($D = K/\gamma_1$, where $K$ is the Frank elastic constant, $\gamma_1$ is the twist viscosity, Methods). $D$ of GO LCs is estimated around $10^{-11}$–$10^{-13}$ m² s⁻¹, orders of magnitude smaller than $10^{-10}$ m² s⁻¹ of small molecular LCs with α 5–100 (Methods)[27]. This teeny $D$ of GO in N phase stems from the giant α of GO and intrinsically causes the retarded relaxation and extreme dynamic stability of reoriented grains. For instance, τ of GO LCs is about $10^6$ s (~30 days) at 0.3 wt.%. By contrast, reoriented grains in traditional LCs (e.g. 4-cyano-4′-pentylbiphenyl) with α around 5–20 fleetly relaxed within a few seconds[17].

**Fundamental $\hat{\mathbf{n}}$ configuration in liquid metacrystals**. Provided the stability of meta-periodicity in GO LCs, we propose SML as a technique to generate highly localized shearing field for $\hat{\mathbf{n}}$

programing and realize LMCs. Correspondingly, the potential of rheological behaviors of colloidal LCs is fully exploited to develop into a lithography for precisely modulating $\hat{\mathbf{n}}$.

Our SML system simply consists of a stage and probes with a moving accuracy of 2 μm (Fig. 2a). The moving probe immersed in GO LCs generates shearing force and rotates GO sheets to align along moving trajectories by overwhelming the elastic torque (Supplementary Fig. 3, Supplementary Fig. 4). By SML, two orientational configurations of GO sheets were fabricated: vertical alignment following the probe movement and pre-aligned horizontal arrangement, identified as the bright and dark domains between crossed polarized light, respectively (Fig. 2b, c, f). Top and section-view polarized optical microscopy (POM) images demonstrate that $\hat{\mathbf{n}}$ undergoes a 180° rotation across mono shearing grain to form a "π wall" (Fig. 2c, f). Controlling probe movement introduced rich topological configurations with intensity covering from −1/2, 1/2 to 1, as well as chiral configurations (Supplementary Fig. 5). After being quenched in liquid nitrogen and freeze–drying, these frozen configurations distinctively revealed that GO sheets are vertically aligned in the bright zone and horizontally arranged in the dark zone (Fig. 2d, e, Supplementary Fig. 6). From section view, π walls penetrate the surface and vertically section the bulk in a regular spacing of ~100 μm (Fig. 2g, h). Other patterns, including dots, cross lines and circles, have the same configuration of GO sheets corresponding to optical observations (Supplementary Fig. 7).

**Mechanism of shearing microlithography**. As a micro-lithography, SML has merits of high resolution, better operability and high throughput in large area, as compared with other electro/magnetic methods[6–9]. Theoretically, the limit of line width ($W$) in GO system was predicted as 4 μm, set by the sum of the radius of probe ($R$) and the boundary layer thickness of the shearing flow. The lower limit of $R$ is determined by the de Gennes-Kleman extrapolation length ($\xi$, estimated as 2 μm, Methods)[28] and the shearing flow conforms to a flow around a circular cylinder[29]. GO solutions had low Reynolds numbers (estimated as $10^{-6} \sim 10^{-8}$, Methods) and kept stable lamellar flow around writing pens. The localized Stokes flows can be described by Navier-Stokes equations (Fig. 3a). We theoretically predicted the relationship between $W$ and $R$, $U$ as

$$W = 2\left(1 + \sqrt{\frac{U}{U_c}}\right)R \qquad (1)$$

where $U$ is the moving velocity and $R$ is the radius of probe $U_c$ stands for the critical velocity to mobilize GO sheets, which is related to the rotation energy barrier of GO sheets. Experimental data are well fitted the function ($W = 2 \times (18.5 \times U^{0.5} + 1) \times 10^{-5}$ when $R$ is fixed at $10^{-5}$ m, and $W = 3.84 \times R$ when $U$ is 2 mm s$^{-1}$, as simulated in Fig. 3b, c, respectively). Provided the independence on the viscosity and fluid density, this clear relationship allows the precise control on $W$ from 10 μm to 2 mm (inset in Fig. 3c, Supplementary Fig. 6), simply by switching probe diameters or adjusting writing speeds.

To experimentally evaluate the resolution limit of SML, a probe with diameter of 5 μm was utilized to construct lamellar periodicity with descending layer spacing down to 10 μm (Fig. 3d–h). The distinct lamellar feature with a spacing of 10 μm demonstrates that the resolution of $W$ by SML reaches 10 μm, which is comparable to the pixel size (dozens of microns) of commercial LC displays. For comparison, existing electric/

magnetic methods are usually taken to fabricate patterns with resolutions ranging from hundreds of microns to several millimeters in GO LCs[6,8].

**Constructing liquid metacrystals with complete 2D symmetry**. SML allows us to group two grains to design LMCs with a library of symmetries (Fig. 4a–g), just through programing the probe movement. In the frame of 2D translation symmetry[30], we constructed LMCs consisting of five crystalline cells, that are oblique, rectangular, rhombic, square and hexagonal. Spreading these cells generated large area meta-periodicity covering five Bravais lattices, as shown in their POM inspections (Fig. 4a, Supplementary Video 1). The power spectra by fast Fourier transform (FFT) reveal the high translational order of LMCs as designed (Fig. 4b)[31].

Beyond the translational symmetry, LMCs extend to quasi-crystals with rotational but not translational symmetry. Guided by Penrose tiling geometry[32], six intriguing cells were designed (Fig. 4d, f). Among these, four cells are pentagonal, octagonal, decagonal and dodecagonal, with 5, 8, 10, and 12 folds rotational symmetry successively, which are usually found in quenched alloys and colloidal assembly[21,33] (Supplementary Video 2). Other two cells are heptagonal and enneagonal, with seven- and nine-folds rotational symmetry, respectively, which are completely artificial. Spreading these six cells harvested LMCs with quasi-crystallinity, exhibited in POM images (Fig. 4d, f). Their power spectra revealed the rotational symmetry as designed (Fig. 4e, g). All these diffraction patterns feature Bragg peaks at low frequency, indicating the order perfectness of designed meta-periodicity (Fig. 4b, e, g). The narrow radial rays demonstrate the long-range orientation order of created LMCs. In a typical LMC with *p6mm* symmetry, we calculated autocorrelation function (ACF) in [100], [010], and [110] crystallographic planes (named as $G_{100}$, $G_{010}$, and $G_{110}$, respectively) to evaluate the structural

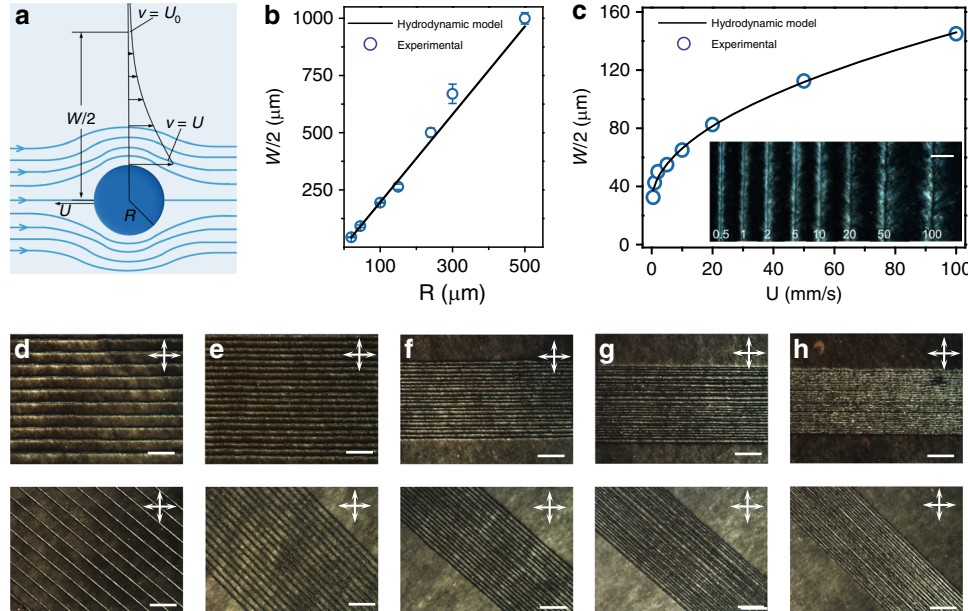

**Fig. 3** Illustration and control mechanism of shearing microlithography. **a** An analogue of the model of flow past a circular cylinder, the width of the line width $W$ is affected by the moving speed $U$ and the radius of the probe $R$ as $W = 2\left(1 + \sqrt{\frac{U}{U_c}}\right)R$ (detailed deduction see Methods). **b, c** Experimental data are well fitted the function ($W = 2 \times (18.5 \times U^{0.5} + 1) \times 10^{-5}$ m (**b**) when $R$ is fixed at $10^{-5}$ m, and $W = 3.84 \times R$ when $U$ is 2 mm s$^{-1}$ (**c**), showing a precise control of SML. Inset in **c** shows the POM images of lines drawn with different velocities, scale bars, 100 μm. **d–h** LMCs with descending layer spacing were fabricated to evaluate the resolution limit of SML. Specifically, a probe with diameter of 5 μm was utilized to construct LMCs with lamellar structures with layer spacing of 100 μm (**d**), 50 μm (**e**), 30 μm (**f**), 20 μm (**g**), 10 μm (**h**). The lamellar feature is still distinguishable at layer spacing of 10 μm, demonstrating the resolution of SML is around 10 μm, Scale bars, 200 μm

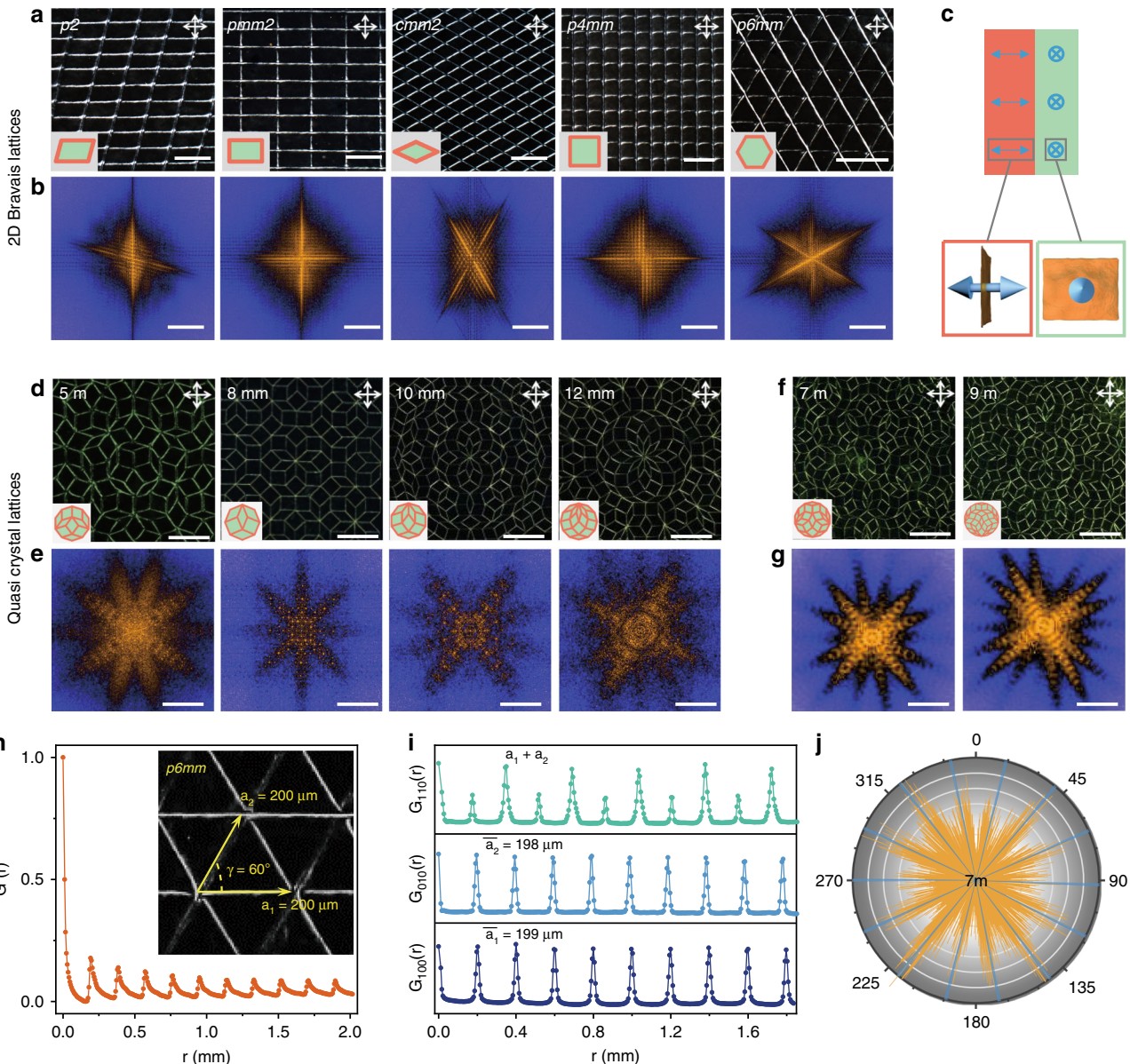

**Fig. 4** Liquid metacrystals with complete 2D symmetry. **a, b** 2D Bravais LMCs. POM images (**a**), Scale bars, 100 μm, and insets are single crystal cells; FFT diffraction patterns (**b**), Scale bars, 1 μm⁻¹. **c** Two basic grains with vertical (orange) and horizontal (green) configurations. **d–g** Quasi-crystalline LMCs. POM images (**d, f**), Scale bars, 500 μm, and insets are rotational repeating cells; FFT diffraction patterns (**e, g**), Scale bars, 0.2 μm⁻¹. **h** The radial ACF of a *p6mm* LMC (inset POM image). **i** ACF calculated for the *p6mm* LMC along [100], [010] and [110] planes. **j** Azimuthal intensity *I(Ψ)* for FFT spectrum of the 7 m LMC. Yellow plots are experimental data, and blue lines are theoretical peaks for ideal 7 m symmetry group

order (Fig. 4h, i). All the experimental ACFs exhibit high consistence with the ideal *p6mm* crystal (Fig. 4i). We also fit the azimuthal intensity *I(Ψ)* in FFT spectra to evaluate the orientation order for a *7m* LMC, and the fitting curve exhibits typical *7m* symmetry characteristic peaks (Fig. 4j). The programmable SML makes it possible for us to construct arbitrary director configurations in 2D space, from natural crystalline and quasi-crystalline lattices to any artificial lattices, which appears as a facile strategy to manipulate LC directors with high complex.

**Extending of diversified liquid metacrystals.** Beyond 2D space, LMC extends to 3D Moiré patterns through a layer-by-layer SML (bilayer honeycomb lattice with a twist angle of 6°), a tetrahedron by adjusting the angle of probes, and color lattices by controlling the distance between π walls (Fig. 5). Notably, besides GO, other colloids with 1D or 2D topology (e.g. cellulose nanocrystals and

zirconium phosphate, Supplementary Fig. 9, Supplementary Fig. 10) and GO polymer composite LC system (Supplementary Fig. 11, Supplementary Fig. 12) were also used to make LMCs successfully. More imaginative, we even fabricated large area $\hat{n}$ maps as artistic paintings and handwritings (Fig. 5b, Supplementary Video 3). Despite the general applicability of SML, it should be pointed out that the stability of LMCs is decided by the relaxation time of LC mesogens and their lateral size. Therefore, the SML might not be a proper candidate for manipulating small molecular LCs or colloids at low concentration (isotropic phase), which usually exhibit rapid relaxation behaviors.

**Relaxation dynamic of liquid metacrystals.** Notably, by tracking the fraction of reorientation area (*f*) and the translational order parameter (*S*) over time in a *p4mm* LMC, we certified that both the reorientation grains themselves and their spatial order relaxed

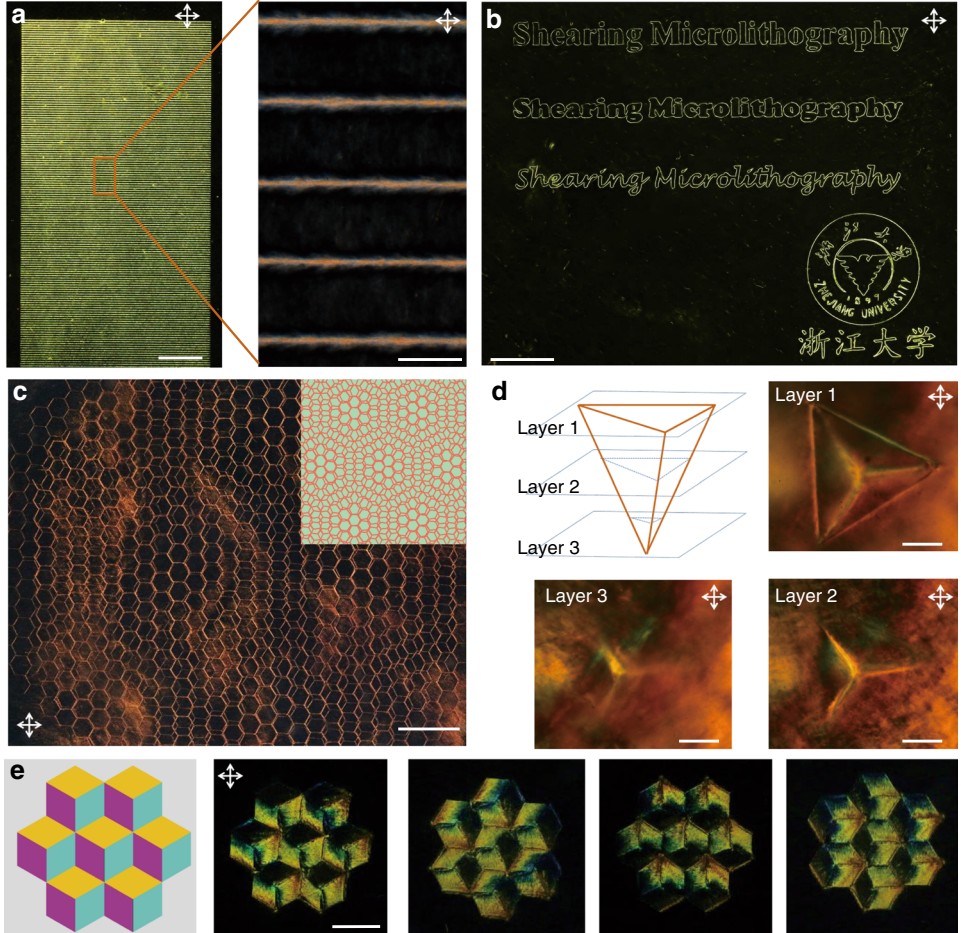

**Fig. 5** Diversified liquid metacrystals constructed by shearing microlithography. **a** POM images of a LMC with lamellar structure with a layer spacing of 200 μm. This large area LMC can be construct within 1 h, showing the high productivity of SML, scale bars, left: 2 cm, right: 200 μm. **b** The freely written patterns of characters with varied fonts, scale bar, 4 mm. **c** LMC with 3D Morie patterns through a layer-by-layer SML, consisting of bilayer 'graphene' lattice with a twist angle of 6°, scale bar, 2 mm. **d** Serial layer scanning of a floating regular tetrahedron program in 3D space at gradually varied focal plane under POM, scale bars, 200 μm. **e** Schematic diagram and POM images of LMCs composed with domain walls orientated in three different directions, which differ from each other in terms of the position of 120° in-plane twist boundary, scale bars, 1 mm

when $c$ is below critical concentration (Fig. 6a, c, Supplementary Fig. 13, Supplementary Fig. 14 and Methods). The relaxation dynamic follows a relationship of $f \propto e^{-\frac{t}{\tau_{1/2}}}$ and $S \propto e^{-\frac{t}{\tau_{1/2}}}$, where $\tau_{1/2}$ is the half relaxation time (Fig. 6a, c, Supplementary Fig. 15), sharing the same Avrami rule of melting kinetics in solid crystals[34,35]. $\tau_{1/2}$ abruptly increases at 0.08 wt.% (Supplementary Table 1), in consistence with $c^*$ for individual grain and I-N transition (Fig. 1b). In the $p4mm$ LMC with $c$ of 0.09 wt.%, the radial ACF gradually lost characteristic peaks with time to 4 h (Fig. 6b), which also declaring the spontaneous fading of translational order.

External vibration accelerates random Brownian movement of GO sheets and drives the melting of lattice (Fig. 6d, Supplementary Fig. 16, Supplementary Fig. 17), in analogy with the thermal-induced melting of crystals[36]. Under forced vibration, GO LMCs with parallel π walls exhibited intense resonance frequency peaks (34 ~ 38 Hz), denoting as melting points. The melting onset frequency ($f_r$, 36 Hz) of LMC is higher than that (30 Hz) of ordinary GO LCs, which means a 44% higher elastic modulus ($E$) of LMC lattices since $E \propto f_r^2$[37]. The melting range (4 Hz) is also narrower than 30 Hz of ordinary LCs. LMCs exhibited an anisotropic melting behavior (Fig. 6d): perpendicular vibration against π walls results in a narrower melting range and higher melting frequency than those of parallel vibration, mainly reflecting the anisotropic strength of crystalline planes.

**Liquid metacrystal based programmable solid materials.** In advance, spatial order in LMC is stable enough to be well preserved after air-drying, thus allowing to program solid materials and rationally design their deformation behavior and overall properties. Under the capillary force during air-drying, horizontal grains forming a compact layer structure, while vertical grains transformed in folded ridge structure (Fig. 7a–d), endowing GO film with meta-periodicity[38,39]. Theoretically, ridges possess higher stiffness than the smooth area in films and guide the condensation of energy into small subsets, resembling the organization of dislocations into crystalline grain boundaries. Correspondingly, we developed a proof-of-concept method of programming "ridgons" to design the deformation behavior together with mechanical and electrical properties of graphene films. In the thin programmed reduced GO (RG) films on elastic polydimethylsiloxane (PDMS) substrate, ridgons prevented the spreading of buckled wrinkles and acted as boundary lines to regionalize local instability in the micro scale (Fig. 7e–i). Smooth RG films generated spreading micro-wrinkles aligned along stretching direction (Fig. 7e, f). While in programmed RG films, ridgons cut off the spreading of wrinkles (Fig. 7g–i).

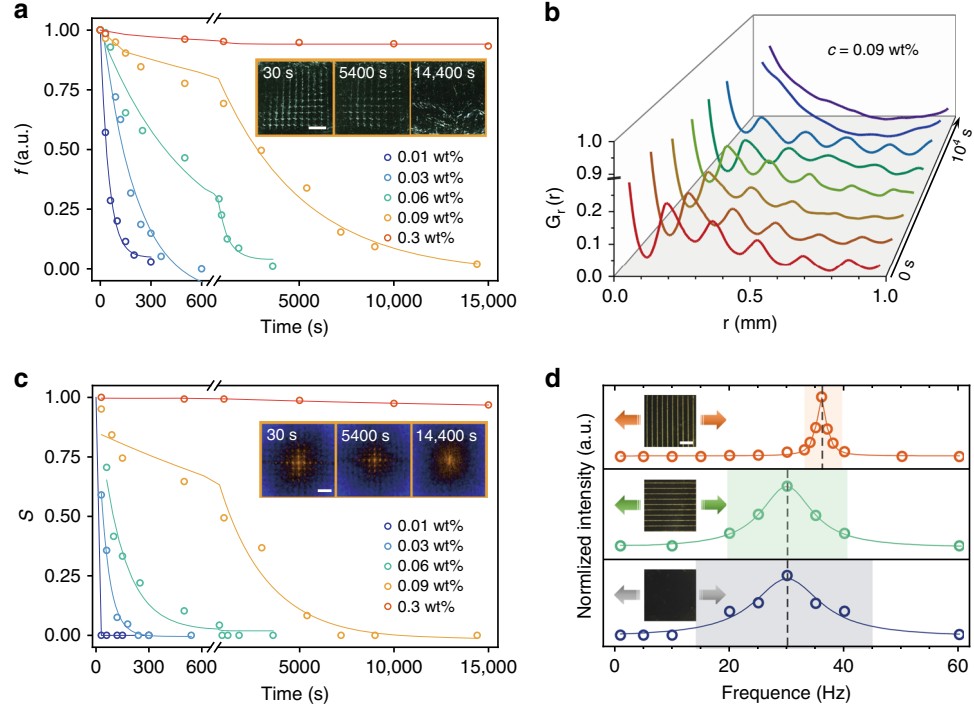

**Fig. 6** Relaxation kinetics and melting behavior of liquid metacrystals. **a** The fraction of reoriented domains ($f$) with time ($t$). Open dots are experimental data and solid lines are fittings by $f = ke^{-\frac{t}{\tau_{1/2}}}$ (See Methods). Insets are POM tracking images of LMC (0.09 wt.%) during the relaxation process, scale bars, 500 μm. **b** RACFs for LMC (0.09 wt.%) during the relaxation process. These characteristic peaks gradually blur with time. **c** $S$ as a function of $t$ and $c$. Solid lines are fittings by $f = ke^{-\frac{t}{\tau_{1/2}}}$ (See Methods). Insets are FFT spectra of LMC (0.09 wt.%) in different period in relaxation, Scale bars, 0.1 μm$^{-1}$. **d** Lattice integrity as a function of vibration frequency. Insets are POM images for LMCs with reoriented domains arrange perpendicular (orange), parallel (green) to the vibration direction and GO LC without meta periodicity (blue); scale bars, 500 μm

We also found that ridgon array enforced the mechanical strength of thick RG bucky papers, dependent on the angle ($\theta$) between ridgons and strain direction (Fig.7j). The longitudinal ridgon array ($\theta = 0°$) exhibited the highest modulus (12.6 GPa), tensile strength (148 MPa) and the longest fracture elongation (3%), outperforming the overall mechanical properties of the control RG papers without patterning, that are 6.7 GPa, 75 MPa, and 2.8% (Fig. 7j), respectively. The programmed ridgons relieved the deterioration tendency of mechanical properties as the area of RG films enlarged. The fracture strength of typical plain RG films decreased to nearly 40% as the size of papers increased (the width of testing papers) from 1 mm to 1 cm. By contrast, programmed RG papers can maintain the tensile strength about 78%, 95% higher than that of smooth papers (Fig. 7k). We attribute this reinforcement mechanism to the regionalized local instability caused by ridge structures (Supplementary Fig. 18). Moreover, the ridgons exhibit a highly anisotropic electrical conductivity (Supplementary Fig. 19). RG papers have conductivities about 6385 S m$^{-1}$ along the ridgons, 740% higher than that in perpendicular direction.

## Discussion

Ever since the foundation of LCs, the precise manipulation of the director field has played the fundamental role for their wide applications. Beyond the molecular level, we created stable LMCs with meta-periodicity in colloidal lyotropic LCs. We took LMC as an extended form of LC and designed it to completely cover a library of symmetries, including Bravais, Penrose tiling lattices in 2D space and diversiform 3D metastructures. Importantly, LMCs were revealed to be long time stable by a dynamic arresting mechanism. The giant anisotropy of colloidal mesogens ($\alpha \sim 10^4$) and their high rotation energy barrier effectively retards the

relaxation of meta-periodicity. As a result, LMC is stable in lyotropic colloidal LCs (e.g. 99.7 wt.% water plus 0.3 wt.% GO) with low viscosity, which favors the localization of shearing field and high resolution of SML. Meta-periodicity in lyotropic LMCs can be intactly converted into meta-structures in their solid materials, exhibiting the ability to design deformation behaviors and overall properties of LC based materials. LMCs with high-level structural order beyond ordinary LCs can serve as a versatile platform to study the fundamental interaction of defects in LCs.

The shear induced alignment of LCs mesogens has been extensively reported and the globe shearing field merely results in a limited spatial manipulation of the LC $\hat{n}$ field, such as blade coating, string, and rotating. In advance, we conceived a localized shearing field to precisely manipulate $\hat{n}$ field and initiated the SML technique. The potential of rheological behaviors of colloidal LCs is fully exploited to develop into a lithography for precisely modulating $\hat{n}$, although the shear-thinning and fluid-induced birefringence have been observed in the early 1930s[24]. Compared with other patterning methods, such as external electric/magnetic fields, laser induced local heating or specific surface anchoring[3–12,17–22], our SML mainly possesses three merits: simple equipment request, programmable designability, and wide compatibility for colloid LC systems. The invented SML extends our limit to precisely design and fabricate desired structures, functionalities, micro- and macroscopic materials, by programing on computers. We believe that the resolution of SML can be further improved from 10 μm at present to nanoscale in the future.

In conclusion, we invented the SML technology to fabricate LMC in colloidal LCs. LMCs have stable artificial meta-periodicity and completely cover a library of symmetries in 2D space and diversiform 3D superstructures. The meta-periodicity of LMCs is dynamically stabilized by the giant molecular size and

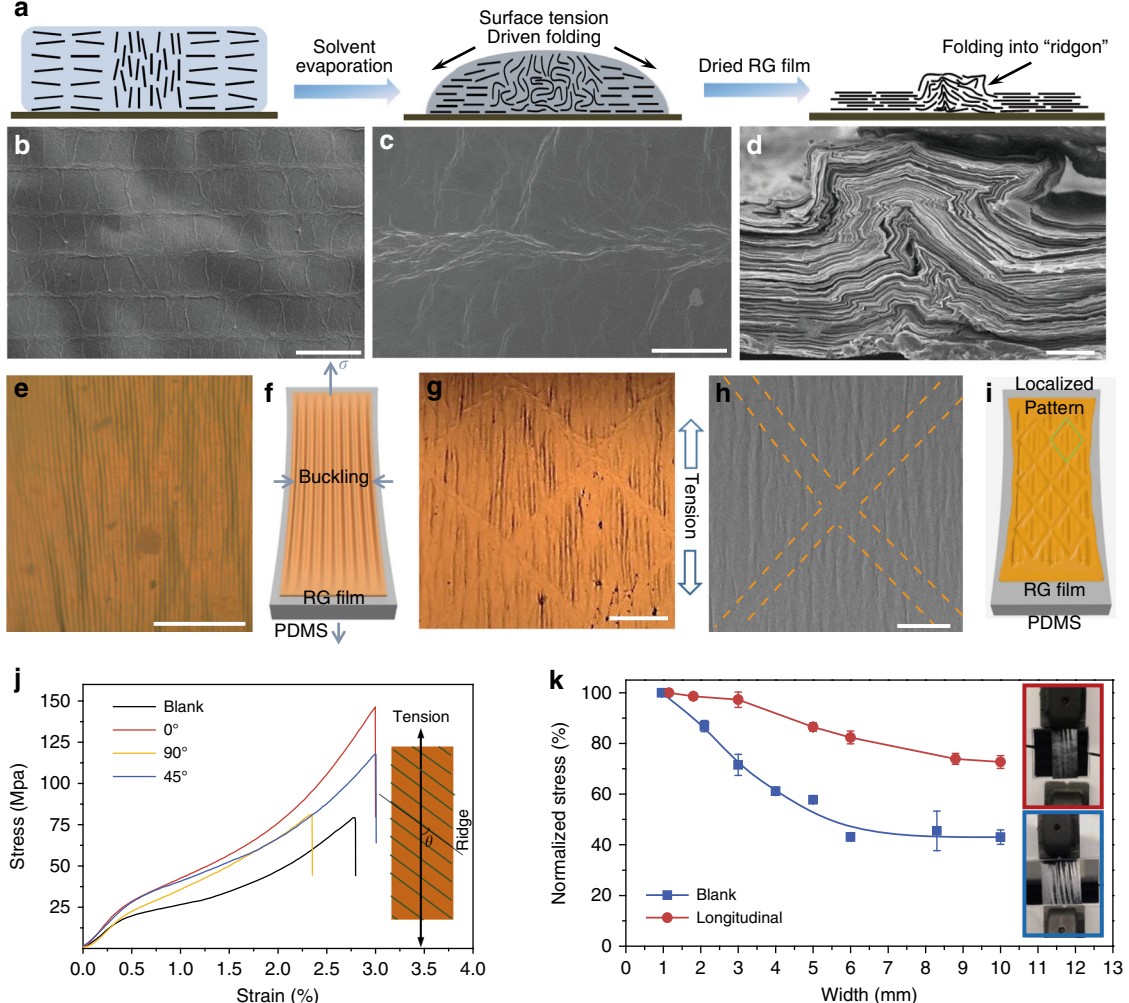

**Fig. 7** Solid films with programed structures obtained from liquid metacrystals. **a** Vertical domains in LMC transformed into "ridges" under capillary force during drying process[36], generating GO films with periodic "ridge" structures. **b–d** SEM images of surface (**b**, **c**) and section (**d**) of ridge on programmed GO film, scale bars, **b**, 400 μm; **c** 50 μm; **d** 5 μm. **e** The optical microscopy image shows smooth GR film on PDMS with throughout wrinkles under uniaxial stress, scale bar, 100 μm. **f** The scheme of wrinkles aligned along strain direction. **g**, **h** The optical microscopy (**g**) and SEM (**h**) images of patterned RG film demonstrate that ridges can prevent the spreading of buckled wrinkles and act as boundaries to regionalize local instability, scale bars, 100 μm. **i** The localization of wrinkles by ridges networks. **j** Strain curves of GO papers with ridges arranged in different angles to the strain direction (inset). **k** Mechanical performances of patterned and smooth GO papers with varying testing width (1–10 mm), the testing length is fixed at 20 mm. The insets with red and blue frames are the snapshots of the strained patterned GO and smooth GO papers, respectively

strong excluded volume repulsion. The SML behaves as a general technique to precisely fabricate meta-structures of colloidal LCs and assembled solid metamaterials. Our work not only extends the structural order of lyotropic LCs but also proposes a facile philosophy to design colloidal LCs and their materials.

## Methods

**Material preparation and characterizations**. Aqueous GO solution (10 mg mL$^{-1}$, lateral width of 10–20 μm) was purchased from Hangzhou Gaoxi Technology Co. Ltd. [www.gaoxitech.com]. Notably, GO sheets process high density of oxygen-containing groups including hydroxyl, epoxy and carboxyl groups. GO sheets are negatively charged in aqueous dispersion with Zeta potential below −30 mV, ensuring the stability of GO solution. Zirconium phosphate powders were purchased from BYK Additives & Instruments. Cellulose nanocrystals were purchased from Aladdin.

Scanning electron microscopy (SEM) images were taken on Hitachi S4800 field emission system. Polarized optical microscopy (POM) and optical microscopy images were taken on ZEISS Axio Scope. A1. FFT was conducted by Image J.

**Phase transition concentration of graphene oxide liquid crystals**. The I-N phase transition of GO solution is explained by Onsager theory based on the

excluded volume effect. As concentration increases, the translational entropy increase compensates the entropy loss of orientation order, driving the system to form thermal dynamic stable liquid crystals spontaneously. The critical transition concentration can be simply predicted as:

$$\varphi = 4\frac{h}{d} \tag{2}$$

where $\varphi$ is the volume fraction, $h$ and $d$ are thickness and lateral size of GO sheets, respectively. In our case, $h$ and $d$ are around 1 nm and 15 μm. Thus the $\varphi$ in our GO LCs is estimated about 0.03 vol.% (0.04 wt.%), which is in agreement of our experimental results (0.06~0.08 wt.%).

**Relaxation dynamics in isotropic phase**. In I phase, GO sheets are randomly distributed, and their rotation would not appreciably alter their neighbors. In this case, the director of GO normally oscillates about equilibrium positions but occasionally crosses an energy barrier to deviate the original equilibrium. This dynamic relaxation behavior can be described by the Arrhenius equation:

$$\tau = A\eta \exp\left(\frac{\Delta E}{k_B T}\right) \tag{3}$$

where $A$ is a constant, $\eta$, $\Delta E$, $k_B$ and $T$ stand for the viscosity of solvent, the height of energy barrier, Boltzmann constant and the temperature, respectively. Previous

theoretic analyses[22,24] have pointed out that the increasing concentration ($c$) shortens the intermolecular distance, resulting in a higher $\Delta E$. Accordingly, the relaxation time $\tau$ is given by:

$$\tau = B_{\mathrm{I}} c^{\rho_{\mathrm{I}}} \qquad (4)$$

where $B_{\mathrm{I}}$, $\rho_{\mathrm{I}}$ are constants related to the kind of colloids.

**Relaxation dynamics in nematic phase.** In N phase, the relaxation time is determined by the dynamics of annihilation of two disclinations with strength of $+1/2$ and $-1/2$. This process is determined by the elastic force $F_{\mathrm{e}}$ and the viscosity drag force $F_{\mathrm{d}}$.

The elastic energy per length of disclination line is

$$W = \pi K (k_1 + k_2)^2 \ln \frac{R}{r_{\mathrm{c}}} - 2\pi K k_1 k_2 \ln \frac{L}{2r_{\mathrm{c}}} \qquad (5)$$

where $K$ is the Frank elastic constant, $k_1$, $k_2$ are strengths of two disclinations, $R$ is the system lateral size, $r_{\mathrm{c}}$ is the radius of the disclination core and $L$ is their neighbor distance. Note that the two disclinations have opposite strength $k_1 = -k_2 = k$, thus the elastic force that attracts each other is given by

$$F_{\mathrm{e}} = -\frac{\partial W}{\partial L} = -2\pi K k^2 / L \qquad (6)$$

To evaluate the viscosity drag force $F_{\mathrm{d}}$, we assume that the defect velocity is too low to cause flow of nematic fluid and the dissipation per unit length ($\Sigma$) is

$$\sum = \gamma_1 \int \left(\frac{\partial \varphi}{\partial t}\right)^2 dx dy \cong \pi \gamma_1 k^2 u^2 \ln \frac{R}{r_{\mathrm{c}}} \qquad (7)$$

where $\gamma_1$ is the twist viscosity and u is the velocity of defects. Therefore, $F_{\mathrm{d}}$ is given by

$$F_{\mathrm{d}} = \frac{\sum}{u} = \pi \gamma_1 k^2 u \ln \frac{R}{r_{\mathrm{c}}} \qquad (8)$$

Notably, deep analysis[40] shows that $F_{\mathrm{d}}$ does not diverge with $R$, because at large distances ($R > R_{\mathrm{c}} \approx 3.6\, K/\gamma_1 u$), the director would re-orientate and the dissipation energy practically vanishes, so that

$$F_d = \pi \gamma_1 k^2 u \ln \frac{3.6}{\mathrm{Er}} \qquad (9)$$

where $Er = \gamma_1 u r_{\mathrm{c}}/K$ is so-called Ericksen number of the problem. Balancing the $F_{\mathrm{e}}$ and $F_{\mathrm{d}}$, we could conclude that

$$\tau \approx C \times \frac{\gamma_1}{K} L^2 \qquad (10)$$

Here, $C$ is a constant. According to previous literatures[24], $K/\gamma_1$ is defined as the diffusivity of the director ($D$):

$$D = {}^K / \gamma_1 \qquad (11)$$

And $D$ is proportional to $c$, that is: $D \propto c^m$, so we can deduce that

$$\tau = B_{\mathrm{N}} c^{\rho_{\mathrm{N}}} \qquad (12)$$

where $B_{\mathrm{N}}$, $\rho_{\mathrm{N}}$ are constants. Typically, for our GO dispersion with $c \sim 0.3{-}2$ wt.%, and $\alpha \sim 15000$, $K$ is estimate around $10^{-10}$ N and $\gamma_1$ is about $10{-}10^3$ Pa s, leading to a $D$ in the range of $10^{-11}{-}10^{-13}$ m$^2$ s$^{-1}$.

In conclusion, whether in isotropic or nematic phases, the relationship between relaxation time ($\tau$) and concentration ($c$) follows a power law. On the basis of our experimental results, in isotropic phase $\tau \propto c^{1.3}$, while in nematic phase $\tau \propto c^{6.7}$. The astonishing increase of exponent originates from the additional energy barrier induced by the excluded volume overlapping in N phase.

**Shearing microlithography process.** Following a conventional film casting protocol[41], GO solution (0.01–0.3 wt. %) was blade casted onto glass substrate to obtain a flat GO liquid layer. Then a microprobe controlled by robotic arm (TH-206H, TIANHAO TECHNIC, China) was immersed into the GO liquid layer and moved according to programmed paths to generate corresponding shearing field. The radius of the microprobe and the moving speed ranged in 5~500 μm and 0.5~100 mm s$^{-1}$ on demand, respectively.

**Control factors of shearing micrography.** The Reynolds number Re can be calculated by Eq. 12:

$$\mathrm{Re} = \frac{\rho U R}{\mu} \qquad (13)$$

For a typical SML process, we chose GO dispersion with a concentration of 5 mg mL$^{-3}$, which exhibit a viscosity ($\mu$) around $10^2$ Pa s and density ($\rho$) around $10^3$ kg m$^{-3}$. Within adequate giving range of parameters ($R$: 5~500 μm, $U$: 0.5~100 mm s$^{-1}$), the Reynolds number Re of GO system is estimated around $10^{-6}{\sim}10^{-8}$.

According to previous reports, when the Re $\rightarrow$ 0, the flow can be treated as Stokes flow and the velocity field can be calculated by Stokes stream function ($\Psi$).

Accordingly, the stream function for flow past a circular cylinder is given by Eq. 14:

$$\Psi = -\frac{U R^2 y}{x^2 + y^2} \qquad (14)$$

Where $U$ is the moving speed of the cylinder, $R$ is the radius of the cylinder, and $x$, $y$ are the coordinate of a given point. Therefore, we can calculate the $u_{\mathrm{x}}$ at point (0, y) by Eq. 15:

$$u_{\mathrm{x}} = \frac{\partial \Psi}{\partial y} = \frac{R^2}{y^2} U \qquad (15)$$

In order to evaluate the width ($W$) of the localized flow field, $W/2$ should be the ordinate of the given point where the velocity is too small to provide enough energy. In practice, we usually define $W/2$ the distance from the center where the velocity equals to a threshold velocity $U_0$. Therefore, we can deduce that:

$$W = 2 \left(1 + \sqrt{\frac{U}{U_{\mathrm{c}}}}\right) R \qquad (16)$$

Where $U_{\mathrm{c}}$ stands for the equivalent critical velocity to rotate GO sheets[42], which is related to the rotation energy barrier of GO sheets. Experimental data are well fitted the function ($W = 2 \times (18.5 \times U^{0.5} + 1) \times 10^{-5}$ m when $R$ is fixed at $10^{-5}$ m, and $W = 3.84 \times R$ when $U$ is 2 mm s$^{-1}$). Provided the independence on the viscosity and fluid density[32,43], this clear relationship allows the precise control on size of domain walls from 10 μm to 4 mm (Fig. 3d–h), simply by switching probe diameters or adjusting writing speed.

**Line width limit of shearing microlithography.** We compared the magnitude of elastic energy and surface anchoring energy to predict the line width limit of SML. As the probe immersed in GO LCs, the orientation of LCs around the probe is determined by the competition between the elastic energy arising from the deformation of LC and the surface-anchoring energy at the LC-probe interface.

Elastic energy ($E_{\mathrm{e}}$) can be written in the well-known Frank form

$$E_{\mathrm{e}} = \int d^3 x \left\{ \frac{K_1}{2} [\hat{\mathbf{n}} (\nabla \cdot \hat{\mathbf{n}})]^2 + \frac{K_2}{2} [\hat{\mathbf{n}} \cdot (\nabla \times \hat{\mathbf{n}})]^2 + \frac{K_3}{2} [(\hat{\mathbf{n}} \cdot \nabla) \hat{\mathbf{n}}]^2 \right\} \qquad (17)$$

where $\hat{\mathbf{n}}$ is the nematic director and $K_1$, $K_2$ and $K_3$ are elastic constants that measure the energy cost for splay, twist and bend deformations, respectively. For simplicity, we do not consider $K_{24}$ and $K_{13}$ terms, which exist in full deformation energy. In the one constant approximation $K = K1 = K_2 = K_3$, the total $E_{\mathrm{e}}$ has the form

$$E_{\mathrm{e}} = \frac{K}{2} \int d^3 x \left[ (\nabla \cdot \hat{\mathbf{n}})^2 + (\nabla \times \hat{\mathbf{n}})^2 \right] \qquad (18)$$

On the surface of the probe, LC sheets tend to align either perpendicular or parallel to the surface. The resulting surface energy ($E_{\mathrm{s}}$) can be written in the Rapini-Popula form

$$E_{\mathrm{s}} = W_{\mathrm{s}} \oint d\sigma (\hat{\mathbf{n}} v)^2 \qquad (19)$$

where $W_{\mathrm{s}}$ is the surface anchoring energy density, v is normal vector to the surface. Notably, the ratio of Frank elastic constant ($K$) and surface anchoring energy density ($W_{\mathrm{s}}$) defines the so-called de Gennes-Kleman extrapolation length

$$\xi = \frac{K}{W_{\mathrm{s}}} \qquad (20)$$

When the radius of the probe $R < \xi$, the deformation is small and the directors can be assumed to be uniform. When $R > \xi$, surface anchoring energy plays a dominant role and the director field follows the surface of probe, which is necessary for the existence of topological defects.

In typical GO LCs, the magnitude of elastic constant ($K$) is $\sim 10^{-10}$ N[44], and the typical scale of the surface anchoring energy density ($W_{\mathrm{s}}$) is around $5 \times 10^{-5}$ J m$^{-2}$. Therefore, the $\xi$ in GO LCs is estimated around 2 μm. In our SML process, the radius of probe is larger than 2 μm (5~500 μm), enabling the guided formation of reoriented grains as probe moves. Adding the boundary layer thickness and the diameter of probe gives the line width of SML about 10 μm.

**Statistics of relaxation dynamic of liquid metacrystals.** Measured POM images in 24-bit RGB color are converted to unsigned 8-bit grayscale and then cropped into a 1024 × 1024 pixels square region. The fraction of light area ($f$) is determined by counting these light pixels and divided by the total number of pixels. We used Image-J to perform a 2D FFT to these greyscale POM images, yielding intensity and phase information. Then, we displayed these FFT intensity images using a color lookup table for clear visualization. To evaluate the translational order ($S$), we calculated the peak area ($A$) in low frequency range for all FFT images, and $S = {}^A / A_0$, where $A_0$ is the peak area in low frequency range of FFT images for corresponding ideal crystal lattices.

**Calculation of autocorrelation function.** To evaluate the spatial order of LMCs, we calculated the autocorrelation functions (ACFs) for LMCs from their 8-bit

grayscale POM images. The general ACF is defined as

$$G(r) = <f(0)f(r)> \qquad (21)$$

where $G(r)$ is the ACF function for given separate distance $r$, and $<>$ represents the ensemble average.

From the discrete intensity data in POM images, we can evaluate ACF as

$$G(r) = \frac{1}{N(M-r)} \sum_{y=1}^{N} \sum_{x=1}^{M-r} I(x+r,y)I(x,y) \qquad (22)$$

where $(x, y)$ is the coordinates in a Cartesian coordinate system with x axis along the given direction (e.g. [100] crystalline plane), $I(x, y)$ stands for the intensity of the point, $M$ and $N$ represent the counts of pixels along $x$ and $y$ axis, respectively. Accordingly, the one dimensional ACF $G_{hlk}(r)$ is calculated with $x$ axis along the direction defined by Miller indices $[hlk]$ from POM images. The radial ACF is obtained by averaging ACFs in all directions.

**Vibration test**. To conduct the forced vibration experiments for LMCs, we fabricated LMCs in 10 mm × 10 mm × 1 mm glass cells, and used an electromagnetic exciter to apply vibrations with frequency ranging from 1~100 Hz. Related POM images were taken before and after a 2-min vibration process to calculate the fraction of light area ($f$) at different vibration frequencies.

## Data availability
The data that support the findings of this study are available on request from the corresponding authors (C.G. or Z.X.).

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

## Acknowledgements
This work is supported by the National Natural Science Foundation of China (Nos. 51533008, 51603183, 51703194, 51803177, 21805242 and 5197030056), National Key R&D Program of China (No. 2016YFA0200200), Fujian Provincial Science and Technology Major Projects (NO. 2018HZ0001-2), Hundred Talents Program of Zhejiang University (188020*194231701/113), Key Research and Development Plan of Zhejiang Province (2018C01049), the Fundamental Research Funds for the Central Universities (No. 2017QNA4036, 2017XZZX001-04), Foundation of National Key Laboratory on Electromagnetic Environment Effects (NO. 614220504030717). We thank Dr. Hua Li for helps on vibration experiments.

## Author contributions
Z.X. and C.G. conceive the study. Y.J. and F.G. designed and carried out the main experiments. Y.J., F.G., W.G., and Z.X. analyzed the data and wrote the main draft of the paper. All authors contributed to the discussion of the results and commented on the paper.

## Additional information

**Competing interests:** The authors declare no competing interests.

