## [Peer Review File · Nature Communications]

Reviewers' comments:

Reviewer #1 (Remarks to the Author):

The authors claimed to have discovered "a new matter state of structural fluids" based on lyotropic liquid crystals (LCs) made of sheets of graphene oxide. In a solution of graphene oxide above a critical concentration, patterns written by micro-lithography remain unchanged for a long period of time. The authors claim that they can create arbitrary periodic superstructures in the liquid crystalline materials. Finally, they characterize the relaxation times of the superstructures, and they show how they can remove the solvent and partially retain such superstructures.

While the LC material created by graphene oxide sheets is certainly interesting, I find most of the claims to be either exaggerated or not substantiated by experiments.

30 years ago it was shown that patterns could be written and stored in high-molecular weight, polymeric liquid crystals (see for example Coles and Simon, Polymer 1985). It is therefore hardly surprising that after shearing a high-molecular weight, very viscous material the system does not relax back. There is one possibly interesting issue, which the paper does not explore, that is the relative role of the material viscosity and that of the creation of topologically protected states.

The superstructures are created by shearing a thin probe across the liquid crystals and "drawing" pi-walls in the sample. The region around the written area is distorted and becomes visible under polarized microscopy due to the birefringence of the medium. Not surprisingly, the distorted region is roughly the size of one graphene oxide sheet (about 10-15 micron large) as can be seen in fig. 2. I understand that these LCs can be then relaxed and erased with high-frequency vibrations. This is rather interesting, but it's very different from the notion of self-organized superstructures in LCs (for example, colloidal assembly in defects in LCs).

Moreover, the images are all shown with polarized microscopy, but it's unclear whether the diffraction pattern is caused by the refractive index gradient near the pi-walls or simply by the scattering due to the creation of ridges on the viscous LC surface - in the latter case, what is the advantage of having a LC system?

Another claim of the paper is that this method has high resolution. The size of what seems to be the defect cores in fig. S3 is more than 20 microns (the nature of the +1 charge defect is in itself debatable), and all the features shown in the papers are hundreds of microns apart, which are much larger distances than those of most LC defect patterns, therefore this claim is puzzling.

The behavior of the system upon drying is interesting, as it shows that the pi-wall defects are retained as the solvent is evaporating. This is not an unexpected or new behavior, as a similar behavior was already shown for example in Yoon et al. PNAS 2013. It is, however, interesting and worth exploring further.

In summary, I think that the very ambitious claims of the authors are not substantiated by the experimental results, therefore I do not recommend the publication of the manuscript in Nature Communications. I would be open to re-assess a modified version of this paper if the authors mitigated their claims and responded to the doubts and issues I've raised.

Reviewer #2 (Remarks to the Author):

This manuscript reports original and very interesting work. The novel SML method presented allows the rapid and inexpensive microscale manufacturing of arbitrary structures based on colloids suspended in a liquid. The manuscript is well written (there are just a few stylistic flaws), methods and evaluation as well as theoretical foundation are presented in a clear way. The method can be of interest for the production of complex photonic crystals, for example. I suggest not to use the term

"super-crystal", because the term "superstructure" is commonly used for a periodic modification of an already periodic structure. In the present case I would speak of a periodic or quasiperiodic "metacrystal", which is defined as a well-ordered structure made of a material that itself consists of atoms/molecules. I recommend acceptance of the manuscript after a minor revision.

Reviewer #3 (Remarks to the Author):

In this manuscript, Y. Jiang et al. reported a very interesting work on making artificial patterns of liquid crystals of graphene oxide by mechanically drawing using a microprobe. They showed a variety of periodic patterns as well as quasicrystalline structures. To me, this technique "shearing microlithography" called by the authors, is kind of a stereolithography technique but using shear force on a liquid crystal medium instead of using laser on a light-sensitive resin solutions. In general, I like the work very much and I think the examples given by the authors show the potential applications of this technique.

I have a few questions regarding on some characterization and technique details, which are listed below:

- 1) One comment is about the generality of this technique. The authors do say that they tried several other solutions, and can obtain patterns using this technique. But I still think more discussions are needed, particularly on the limitations of this technique as well as the comparison with other techniques. This will help readers to understand the technique better.
- 2) More specific details of the system are needed. For example, how the graphene oxide sheets stabilized? Are they charged? In the aqueous solution, besides particles, is just pure water or are there some additives too? The authors do indicate that the system that they used has been reported in their previous paper, but it will be much more convenient for readers to get such information directly in this manuscript rather than to refer to other paper.
- 3) One specific technical question is on the relaxation time. In figure 1, the authors measure the relaxation time by timing the annihilation of a pair of defects. To me, this is a qualitative characterization. It will be affected by the distance between defects, and also can not isolate the contribution of particles translation from that of particles rotation. Typically, the relaxation time can be measured from the auto correlation function or intermediate scattering function by fitting with an exponential decay function. The particles used are pretty big (tens micron), so it would be straightforward to track the motion of particles using optical microscope and get such auto correlation functions.
Relating to this, for the relaxation dynamics of LSC, the data points at high concentrations (Fig. 6a,c) clearly don't follow a simple exponential decay, so it is not clear to me what the meaning of the tau obtained by exponential fitting at those concentrations is. In fact, it looks that the data at the high concentrations in Fig. 6a,c showed two step relaxation processes, indicating that there might be dynamic heterogeneity in the system.
- 4) How do the authors erase patterns? The authors showed that they can erase and rewrite (Fig. S16), but I didn't find the text to explain how they did.
- 5) Line 276-277, the authors stated that "It should be pointed out that ΔE varies proportionally with the concentration c ." However, the concentration c inversely change with the cube of separation distance, but there is no information on how the energy barrier ΔE varies with the distance. Without this information, it is hard to tell the above statement true or not.

Minor thing:

Line 350: Should D be W?

In summary, I will recommend this manuscript to be published after and only after the authors have properly addressed the above comments.

Point-to-point Responses

Reviewer #1

General Comments: The authors claimed to have discovered "a new matter state of structural fluids" based on lyotropic liquid crystals (LCs) made of sheets of graphene oxide. In a solution of graphene oxide above a critical concentration, patterns written by micro-lithography remain unchanged for a long period of time. The authors claim that they can create arbitrary periodic superstructures in the liquid crystalline materials. Finally, they characterize the relaxation times of the superstructures, and they show how they can remove the solvent and partially retain such superstructures. While the LC material created by graphene oxide sheets is certainly interesting, I find most of the claims to be either exaggerated or not substantiated by experiments. 30 years ago it was shown that patterns could be written and stored in high-molecular weight, polymeric liquid crystals (see for example Coles and Simon, Polymer 1985). It is therefore hardly surprising that after shearing a high-molecular weight, very viscous material the system does not relax back. There is one possibly interesting issue, which the paper does not explore, that is the relative role of the material viscosity and that of the creation of topologically protected states.

In summary, I think that the very ambitious claims of the authors are not substantiated by the experimental results, therefore I do not recommend the publication of the manuscript in Nature Communications. I would be open to re-assess a modified version of this paper if the authors mitigated their claims and responded to the doubts and issues I've raised.

Response: Thanks for your comment and constructive suggestions. To remove the possible misunderstanding and misconception, we have mitigated the ambitious claims to make our paper more accurate. To relieve your concerns, we also conducted a series of experiments in our revised manuscript, including rheological tests (Fig. S14), LMCs with higher resolution (Fig. 4), a discussion on the resolution of SML (Fig. 3), control experiments (Fig. S19) and related descriptions. We also give more detailed discussions

on the experiments and limitation of SML (Line 250-278), and cited more related literatures to support our demonstration. We hope that our improved manuscript can address your concerns.

After pondering your concern, we still deeply believe that the colloidal liquid metacrystals together with the shearing microlithography presents a meaningful step toward precise manipulation of liquid crystalline directors. This work not only extends the structural order of lyotropic LCs to a higher level, but also offers a convenient tool to design colloidal LCs and their materials.

- (1) The presented colloidal liquid metacrystal (LMC, renamed according to Reviewer 2's suggestion) is an extended form of colloidal LCs with high-level structural order. The pioneering work by Coles and Simon (Coles and Simon, *Polymer* 1985) explored to generate stable director pattern by external electric field in high viscous polymeric thermotropic LC. Their work implied the possibility to create metastructures in LCs, whereas, we make this deduction come true by the shearing microlithography in low viscous lyotropic LCs that contain ~99.7 w.t.% water and only 0.3 w.t.% GO solute. We demonstrated that LMC can represent a new view to manipulate directors of colloidal LCs and a systematic guidance to design their materials structures.
- (2) In this work, we exhibit the stable metaperiodicity in low viscosity lyotropic LCs. GO LCs have a much lower viscosity (~0.1-10 Pa·s at 0.3 wt.%, comparable to glycerin), distinct from the high viscosity ($10^3\sim 10^5$ Pa·s) of thermotropic polymer LC. In this low viscosity case of GO dispersions, the stability of metastructure was mainly attributed to the giant size of GO sheets and excluded volume of LC. Therefore, the stability mechanism and physical significance are quite different between our work and previous case.
- (3) Thanks for your suggestion. We have further tested the viscosity behaviors of GO dispersions (Fig. S14). Actually, the viscosity of our GO LC case is quite low since the dispersion contains ~99.7 w.t.% water and only 0.3 w.t.% GO solute. Therefore, viscosity might not be the main factor that stabilize LMC.

Comment 1: The superstructures are created by shearing a thin probe across the liquid crystals and "drawing" pi-walls in the sample. The region around the written area is distorted and becomes visible under polarized microscopy due to the birefringence of the medium. Not surprisingly, the distorted region is roughly the size of one graphene oxide sheet (about 10-15 micron large) as can be seen in fig. 2. I understand that these LCs can be then relaxed and erased with high-frequency vibrations. This is rather interesting, but it's very different from the notion of self-organized superstructures in LCs (for example, colloidal assembly in defects in LCs). Moreover, the images are all shown with polarized microscopy, but it's unclear whether the diffraction pattern is caused by the refractive index gradient near the pi-walls or simply by the scattering due to the creation of ridges on the viscous LC surface - in the latter case, what is the advantage of having a LC system?

Response: Thanks for your comment and suggestions.

- (1) To avoid the misunderstanding of the superstructure by self-organization, we have renamed the "super-crystal" to "metacrystal", which will be more accurate in the artificial structures.
- (2) We thought that the diffraction pattern is caused by the refractive index gradient near the pi-wall, reasons are listed as follow: (i) the concentration of GO LCs we utilized to fabricate superstructure is around 0.3~0.5 *w.t.*%, this low solid content ensure the high flowability and low viscosity of the GO LCs, as displayed in Fig. S14 in the revised manuscript, therefore the surface heals up immediately after the shearing, leaving no remnant surface wrinkle. In fact, ridges were observed after and only after the evaporation of water in GO LCs. In liquid state, no ridges were observed on the surface of GO LCs. (ii) We have applied our SML in a few viscous polymer aqueous solution as control experiments, as exhibited in Fig. S19 in the revised manuscript. No related patterns were observed after the SML process. In a short word, our further experiments demonstrate that the diffraction pattern is caused by the refractive index gradient near the pi-wall.

Comment 2: Another claim of the paper is that this method has high resolution. The size of what seems to be the defect cores in fig. S3 is more than 20 microns (the nature of the +1 charge defect is in itself debatable), and all the features shown in the papers are hundreds of microns apart, which are much larger distances than those of most LC defect patterns, therefore this claim is puzzling.

Response: Thanks for your comments.

We theoretically analyzed that the theoretical limit of our SML in GO system is around 10 microns. All the figures in our paper was conducted by a probe with a diameter of 20 microns in order to improve definition of pattern features under optical microscopy. To our knowledge, in GO LC system, where the lateral size of GO sheet is around 1~100 microns, the typical size of defect is about tens of microns. On the basis of results provided by previous literatures about orientation patterns in GO LCs (*Nat. Mater.* **13**, 394-399 (2014). *Adv. Mater.* **29**, 1604453 (2017). *Soft Matter* **7**, 11154-11159 (2011).), we pondered that our method processes high resolution in GO LC systems. We have added detailed discussions in the revised manuscript (line 141-148).

In addition, to address your concern, we have fabricated LSCs with higher resolutions to replace the original figures in our revised manuscript (Fig.4) and fabricated LMCs with decreasing layer spacing down to 10 μm (Fig. 3d-h) to evaluate the resolution limit of our SML.

Comment 3: The behavior of the system upon drying is interesting, as it shows that the pi-wall defects are retained as the solvent is evaporating. This is not an unexpected or new behavior, as a similar behavior was already shown for example in Yoon et al. *PNAS* 2013. It is, however, interesting and worth exploring further.

Response: Thanks for your insight reminder.

In the paper *PNAS* 2013, Yoon et al. used a thermal sublimation strategy to retain the defect structures against surface tension to investigate the internal structural features of 3D samples. In our case, the periodic pi-wall structures were transformed into ridge-

like structures under the capillary force during the air-drying process. We have cited more related literatures in our revised manuscript.

Moreover, after the simple trying in this paper, we think that the pi-wall ridges possibly behave as twin-boundaries in 2D laminated structures (*Nat. Commun.* **9**, 3597 (2018).) and the SML can be used as a general method to construct designed boundaries in layered materials. Following this work, we are taking a systematic research to explore this subject, hoping to unveil new facets of familiar laminated materials.

Reviewer #2

General Comments: This manuscript reports original and very interesting work. The novel SML method presented allows the rapid and inexpensive microscale manufacturing of arbitrary structures based on colloids suspended in a liquid. The manuscript is well written (there are just a few stylistic flaws), methods and evaluation as well as theoretical foundation are presented in a clear way. The method can be of interest for the production of complex photonic crystals, for example. I suggest not to use the term “super-crystal”, because the term “superstructure” is commonly used for a periodic modification of an already periodic structure. In the present case I would speak of a periodic or quasiperiodic “metacrystal”, which is defined as a well-ordered structure made of a material that itself consists of atoms/molecules. I recommend acceptance of the manuscript after a minor revision.

Response: Thanks for your highly appreciated comment on our work. We have replaced the term “super-crystal” with “metacrystal” according to the reviewer’s constructive suggestion, which is indeed more decent for our case.

Reviewer #3

General Comments: In this manuscript, Y. Jiang et al. reported a very interesting work on making artificial patterns of liquid crystals of graphene oxide by mechanically

drawing using a microprobe. They showed a variety of periodic patterns as well as quasicrystalline structures. To me, this technique “shearing microlithography” called by the authors, is kind of a stereolithography technique but using shear force on a liquid crystal medium instead of using laser on a light-sensitive resin solution. In general, I like the work very much and I think the examples given by the authors show the potential applications of this technique. I have a few questions regarding on some characterization and technique details, which are listed below:

In summary, I will recommend this manuscript to be published after and only after the authors have properly addressed the above comments.

Response: Thanks for your appreciated comment and constructive suggestions.

Comment 1: One comment is about the generality of this technique. The authors do say that they tried several other solutions, and can obtain patterns using this technique. But I still think more discussions are needed, particularly on the limitations of this technique as well as the comparison with other techniques. This will help readers to understand the technique better.

Response: Thanks for the reviewer’s constructive advice. We have added more discussions on the generality as well as the limitations in our revised manuscript (Line 188-192, Line 265-278).

Comment 2: More specific details of the system are needed. For example, how the graphene oxide sheets stabilized? Are they charged? In the aqueous solution, besides particles, is just pure water or are there some additives too? The authors do indicate that the system that they used has been reported in their previous paper, but it will be much more convenient for readers to get such information directly in this manuscript rather than to refer to other paper.

Response: Thanks for the reviewer’s comment. Graphene oxide (GO), synthesized by oxidation of graphite, is the most common precursor of graphene with high density of oxygen-containing groups including hydroxyl, epoxy and carboxyl groups. Therefore,

GO sheets are negatively charged in aqueous dispersion with Zeta potential below -30 mV, ensuring the stability of GO solution (*Adv. Mater.* **29**, 1606794, 2017; *Soft Matter* **7**, 11154-11159, 2011). The GO dispersion contains neat GO and pure water, without any additives. In addition, we have provided more details about graphene oxide dispersion in the revised manuscript. (Line 291-294)

Comment 3: One specific technical question is on the relaxation time. In figure 1, the authors measure the relaxation time by timing the annihilation of a pair of defects. To me, this is a qualitative characterization. It will be affected by the distance between defects, and also can not isolate the contribution of particles translation from that of particles rotation. Typically, the relaxation time can be measured from the auto correlation function or intermediate scattering function by fitting with an exponential decay function. The particles used are pretty big (tens micron), so it would be straightforward to track the motion of particles using optical microscope and get such auto correlation functions.

Relating to this, for the relaxation dynamics of LSC, the data points at high concentrations (Fig. 6a, c) clearly don't follow a simple exponential decay, so it is not clear to me what the meaning of the tau obtained by exponential fitting at those concentrations is. In fact, it looks that the data at the high concentrations in Fig. 6a,c showed two step relaxation processes, indicating that there might be dynamic heterogeneity in the system.

Response: Thanks for your kind advice. Theoretically, the optical absorption of single graphene sheet is about 2.3%, and the optical absorption of graphene oxide (GO) is much weaker than graphene (*Nat. Nanotechnol.* **2008**, *3*, 101–105. *J. Am. Chem. Soc.* **2010**, *132*, 260–267), therefore, it has been a long-lasting challenge to visualize single GO sheet under optical microscope. Limited by these technique challenge, we are compelled to measure the relaxation time by timing the annihilation of a pair of defects.

As to the relaxation dynamic in Fig. 6a, c, the data points actually follow a typical exponential decay. We have broken the time axis to makes data at the beginning clearer,

which misleading these data to appear a two-step relaxation. In order to eliminate further misunderstanding, we have provided the original figures without breaking in the revised manuscript (Fig. S9), which exhibit a typical exponential decay assuredly.

Comment 4: How do the authors erase patterns? The authors showed that they can erase and rewrite (Fig. S16), but I didn't find the text to explain how they did.

Response: Patterns were erased by manual vibrations, we have added the details of the erase and rewrite process in the caption of Fig. S18 in revised manuscript.

Comment 5: Line 276-277, the authors stated that “It should be pointed out that delta E varies proportionally with the concentration c .” However, the concentration c inversely changes with the cube of separation distance, but there is no information on how the energy barrier delta E varies with the distance. Without this information, it is hard to tell the above statement true or not.

Response: Thanks for the reviewer's comment. We made the statement that “delta E varies proportionally with the concentration c .” to qualitatively express that delta E increases with increasing concentration, which was based on the results of previous literatures: Langmuir, I. *J. Chem. Phys.* **6**, 873-896 (1938). Onsager, L. *Ann. N. Y. Acad. Sci.* **51**, 627-659 (1949). In order to reduce misunderstanding, we have correct this statement in our revised manuscript as fellow: Previous theoretic analyses^{15,17} have pointed out that the increasing concentration (c) shortens the intermolecular distance, resulting a higher ΔE . Accordingly, the relaxation time τ is given by: $\tau = B_1 c^{\rho_1}$. (Line 318-321)

Comment 6: Minor thing: Line 350: Should D be W?

Response: Thanks for the reviewer's kind reminder, we have corrected the mistake.

REVIEWERS' COMMENTS:

Reviewer #1 (Remarks to the Author):

The authors of the manuscript *Artificial Colloidal Liquid Metacrystals*, by Jiang et al, have addressed all of my previous comments. The authors have mitigated some of their claims, as I suggested, and their comments and additional figures have dissipated several doubts that I had on this work. I think this work is now acceptable for publication, but there are minor revisions I recommend here below.

1) I appreciate the study of the viscosity in figure S14. Could the authors indicate, either in the caption or in the figure, what is the range of shear rates corresponding to the velocity of the writing tip?

2) On line 165-166 the author mention that they can pattern quasi-crystal with 7 and 9 fold that have never been found so far. In this system any 2D pattern is accessible, as it is created by writing. Micro-pillars or other topographical features that are able to align LC can be patterned with a 7 and 9 fold symmetry, and that also would induce a 7-fold or 9-fold defect array. Yet none of these structures would advance the big search for quasi-crystalline self-assembled (or directed-assembled) structures. I think the authors should clarify this point more.

3) line 181: I believe the authors refer to Moire' pattern, not Morie

4) How was anchoring strength estimated on line 424? The work by Collings (Collings et al. *Liquid crystal* 44, 1165) is one of the few works, to the best of my knowledge, on anchoring of lyotropics and the anchoring strength is two orders of magnitude lower. Could the authors clarify this point?

Reviewer #2 (Remarks to the Author):

The manuscript has been improved by the revisions and should be published as it is.

Reviewer #3 (Remarks to the Author):

In the revised manuscript, the authors have answered my concerns properly including adding more discussions on the generality of the technique, and more details on the experiments. So I would like to recommend the current manuscript to be published in *Nature Communications*.

Point-to-point Responses

Reviewer #1

General Comments: The authors of the manuscript Artificial Colloidal Liquid Metacrystals, by Jiang et al, have addressed all of my previous comments. The authors have mitigated some of their claims, as I suggested, and their comments and additional figures have dissipated several doubts that I had on this work. I think this work is now acceptable for publication, but there are minor revisions I recommend here below.

Response: Thanks for your pertinent comments on our work. We have revised our manuscript to address your concerns and the point-to-point response is listed below. Sincerely thanks for your constructive suggestions on improving our manuscript.

Comment 1: I appreciate the study of the viscosity in figure S14. Could the authors indicate, either in the caption or in the figure, what is the range of shear rates corresponding to the velocity of the writing tip?

Response: Thanks for your comment. We have provided the corresponding shear rates of the probe in the related caption in the revised manuscript. In a typical SML process, the moving speed of probe is around $1 - 100 \text{ mm s}^{-1}$, and the diameter of the probe is around $5 - 500 \text{ }\mu\text{m}$. Therefore, the corresponding shear rate of the probe is around $10^0 - 10^4 \text{ s}^{-1}$.

Comment 2: On line 165-166 the author mention that they can pattern quasi-crystal with 7 and 9 fold that have never been found so far. In this system any 2D pattern is accessible, as it is created by writing. Micro-pillars or other topographical features that are able to align LC can be patterned with a 7 and 9 fold symmetry, and that also would induce a 7-fold or 9-fold defect array. Yet none of these structures would advance the big search for quasi-crystalline self-assembled (or directed-assembled) structures. I think the authors should clarify this point more.

Response: Thanks for your comments. The versatility of the presented SML allows the manipulation of LC directors with high complex, just as you comment by “writing”. In

order to mitigate the potential misunderstanding of our method with the previous self-assembled quasicrystals, we have modified the related statement in our revised manuscript (Line 160, Line 171-175) by merely exhibiting the versatility of our SML method.

Comment 3: line 181: I believe the authors refer to Moire' pattern, not Morie

Response: We have corrected this spelling mistake.

Comment 4: How was anchoring strength estimated on line 424? The work by Collings (Collings et al. Liquid crystal 44, 1165) is one of the few works, to the best of my knowledge, on anchoring of lyotropics and the anchoring strength is two orders of magnitude lower. Could the authors clarify this point?

Response: The anchoring strength for GO LCs we used in our manuscript is derived from previous literatures on GO lyotropic LCs (*Physical Review Letters*, **2005**, 95(15): 157801. *Nano Letters*, **2010**, 10(4): 1347-1353. *Soft Matter*, **2011**, 7(23): 11154-11159.). We reason that the ultrahigh anchoring strength of GO LCs, two orders of magnitude higher than small molecular LCs, is caused by ultralarge aspect ratio of GO sheets together with their rich functionalities that have strong affinity to substrate.

Reviewer #2

General Comments: The manuscript has been improved by the revisions and should be published as it is.

Response: Thanks for your highly appreciated comment on our work.

Reviewer #3

General Comments: In the revised manuscript, the authors have answered my concerns properly including adding more discussions on the generality of the technique,

and more details on the experiments. So I would like to recommend the current manuscript to be published in Nature Communications.

Response: Thanks for your appreciated comment.